# Capturing Plume Rise and Dispersion with a Coupled Large-Eddy Simulation: Case Study of a Prescribed Burn

**Nadya Moisseeva** *  **and Roland Stull**

Department of Earth, Ocean and Atmospheric Sciences, The University of British Columbia,
Vancouver, BC V6T 1Z4, Canada; rstull@eoas.ubc.ca
* Correspondence: nmoisseeva@eoas.ubc.ca

**Abstract:** Current understanding of the buoyant rise and subsequent dispersion of smoke due to wildfires has been limited by the complexity of interactions between fire behavior and atmospheric conditions, as well as the uncertainty in model evaluation data. To assess the feasibility of using numerical models to address this knowledge gap, we designed a large-eddy simulation of a real-life prescribed burn using a coupled semi-emperical fire–atmosphere model. We used observational data to evaluate the simulated smoke plume, as well as to identify sources of model biases. The results suggest that the rise and dispersion of fire emissions are reasonably captured by the model, subject to accurate surface thermal forcing and relatively steady atmospheric conditions. Overall, encouraging model performance and the high level of detail offered by simulated data may help inform future smoke plume modeling work, plume-rise parameterizations and field experiment designs.

**Keywords:** wildfire plume rise; smoke modeling; large eddy simulation; emissions dispersion; WRF-SFIRE; RxCADRE

## 1. Introduction

Wildland fires cover a broad range of spatiotemporal scales and are shaped by the complex interaction of fuel, terrain, and meteorological conditions. While scientific understanding of wildland fires and associated smoke plumes are central to the successful mitigation of negative air-quality impacts, the complex and highly dynamic nature of fires presents a challenge for modeling. Existing smoke plume prediction models span a vast range of complexity from simple empirical relations to the more recent coupled fire–atmosphere numerical approaches. Often the choice of model is dictated by the context of its application, subject to the trade-off between fidelity and timely execution.

Large eddy simulation (LES) is a method that uses computational fluid dynamics at a very fine spatial and temporal resolution to simulate a wide range of scales of atmospheric motions down to the size of large turbulent eddies. The Weather Research and Forecasting Model, combined with a semi-empirical fire-spread algorithm (WRF-SFIRE), allows two-way coupling between an LES and a fire behavior model [1–3]. Several studies have examined the ability of WRF-SFIRE to capture the ground-spread behavior of a fire line, near-surface temperatures and winds [1,4,5]. Large-scale simulations of two real fires were carried out by Kochanski et al. [6], comparing modeled plume tops with satellite data. To the authors' best knowledge, very limited consideration has been given to assessing the ability of WRF-SFIRE to simulate wildfire smoke plume dynamics, and vertical rise and distribution of emissions on a local scale. This is the central motivation for this study.

As noted by Mallia et al. [7], there is a general lack of research focusing on modeling the vertical distribution of smoke emissions as a result of wildfires. This knowledge gap can, in part, be explained

by the difficulty of constraining potential sources of error in both inputs and models themselves. Until recently, evaluation of coupled fire–atmosphere models required a combination of studies, as no dataset was complete enough to rigorously constrain the problem [1]. Comprehensive field observations were needed to better our understanding of the interactions between fuels, fire behavior and meteorology.

In response, the Prescribed Fire Combustion and Atmospheric Dynamics Research Experiment (RxCADRE) was designed to address this critical research need [8]. The project brought together researchers from a wide range of disciplines to collect data on fuel, meteorology, fire behavior, energy, smoke emissions and fire effects. Simultaneous measurement of multiple fire aspects on the same prescribed burns provided a detailed model evaluation dataset, while also capturing the effects of fire–atmosphere coupling [9]. Data from this comprehensive experiment offers a unique opportunity to assess the accuracy WRF-SFIRE simulated plume rise and dynamics.

In addition, the modeling work we present may help inform future observational studies by identifying key aspects of experimental design. Note that the focus of this work is the evaluation of the LES ability to capture the atmospheric response to a simulated fire of known bulk properties, rather than the fire behavior itself. Effectively, the work aims to validate the relationship between the simulated surface forcing due to a fire and the resultant turbulent convection.

The findings are likely to be of interest for atmospheric and air-quality modelers, as detailed measurements of wildfire smoke plumes are scarce. "Synthetic" plume data from an LES would provide researchers with an alternative resource for validating their models. Therefore, the broad goal of this work is to assess the utility of WRF-SFIRE for improving plume rise and dispersion parameterizations.

## 2. Methods

### 2.1. Observational Data

The RxCADRE campaign consisted of 10 operational and 6 small replicate prescribed fires. Collected data are accessible via a US Forest Service online repository, as referenced below. Smoke dispersion and emissions measurements are available for three large fires: L1G and L2G grass fires and L2F sub-forest canopy surface fire. For the purpose of model evaluation, we selected L2G (10 November 2012) for our case study, based on its reported uniformity and consistency of flame propagation [10]. Figure 1 shows a sample snapshot of the burn plot during the ignition. The overall meteorological conditions and instrumental design of the L2G experimental burn are described in detail in [9]. The individual datasets obtained from the US Forest Service online archive used for this study are summarized below.

Georeferencing data, including plot location and burn perimeters, are available from Hudak and Bright [11]. Analysis of fire rate of spread (ROS) and intensity as well as a detailed description of three Highly Instrumented Plots (HIPs) used to produce the estimates can be found in [10]. Locations of HIPs are available from Hudak et al. [12]. HIP1, used for this evaluation, is shown in Figure 1. Near-surface wind and temperature sonic anemometer time series for in-situ and background locations are available from Seto and Clements [13,14]. Ignitions timing and locations were obtained from field-grade GPS units, mounted on-board firing vehicles [15]. Fuel data used for this evaluation study included photographs of pre-burn samples, as well as measurements of size, loading and moisture content of species groups. Data collection methodology is detailed in [16]. Dispersion and emissions measurements included volume-mixing ratio of $CO_2$, CO, $CH_4$, and water vapor at a rate of 2 s, obtained from aircraft-mounted sensors [17]. The georeferenced data consisted of horizontal transects at multiple elevations, as well as "corkscrew" and "parking garage" flight profiles.

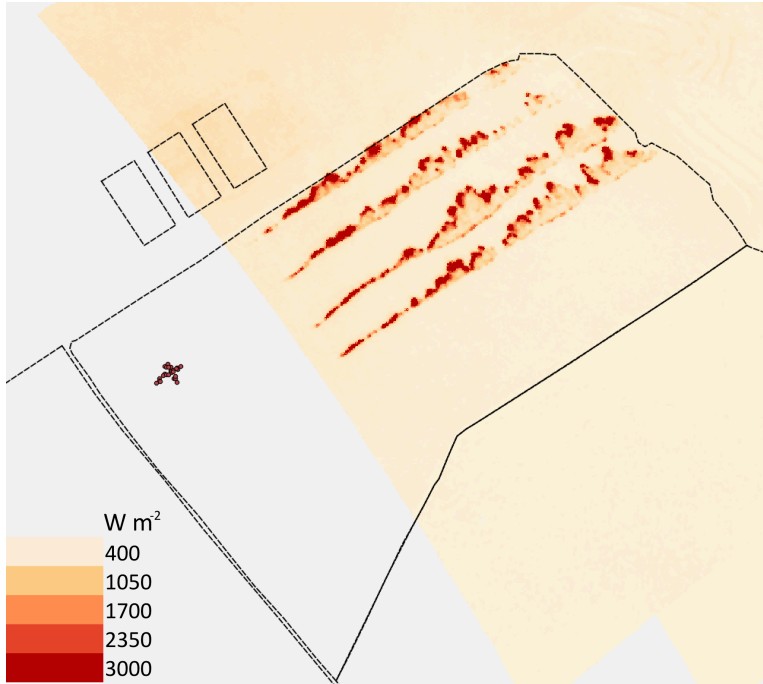

**Figure 1.** Long wave infra-red (LWIR) image of L2G lot during ignition (12:32:02 CST) with dashed black lines denoting burn perimeters. Red scatter points correspond to Highly Instrumented Plot (HIP) #1 fire behavior packages (FBP), each containing a system of airflow, temperature and energy sensors.

## 2.2. Numerical Setup

WRF-SFIRE [3] was configured in idealized LES mode. One of the primary advantages of using this model is that it allows for two-way coupling between the fire and the atmosphere. While WRF-SFIRE does not model combustion directly, the spread and intensity of the fire are parameterized using a semi-empirical approach. The latent heat flux is computed based on the fuel consumption and stoichiometric combustion of cellulose. Heat and moisture fluxes from the simulated burn provide forcing to the atmosphere, which in turn influences fire behavior.

A 10.4 km $\times$ 14 km domain with 40 m horizontal grid spacing, 3000 m model top and 51 hyperbolically stretched vertical levels was initialized using the 10:00 CST (16:00 UTC) sounding [9]. While this may appear to be a shallow domain compared to mesoscale ("Real") WRF simulations, the choice is substantially higher than that found in existing published WRF-SFIRE evaluations [1,4,18]. Five lowest model grid centers were located at approximately 8 m, 24 m , 42 m, 60 m and 80 m above ground level (AGL). The simulation was allowed to spin up for 2 h 23 min prior to ignition at $\sim$12:23 CST (time varied slightly for different fire lines). To aid the formation of buoyancy-driven ambient background turbulence typical for a daytime boundary layer (BL), a lower-boundary surface thermal flux (tke_heat_flux) was imposed. The value was estimated from the sonic anemometer time series of vertical wind velocity and temperature over the time period leading up to ignition. As shown in Figure 2, based on the measurements, the ambient background surface heat flux remained fairly constant over the entire spin-up period. Hence, the lower-boundary surface forcing was idealized for the LES simulation as being uniform in space and constant in time. We used full surface initialization (sfc_full_init =.true.), with the lower boundary moisture flux and surface roughness characteristics set to standard USGS values for "Grassland" land use category.

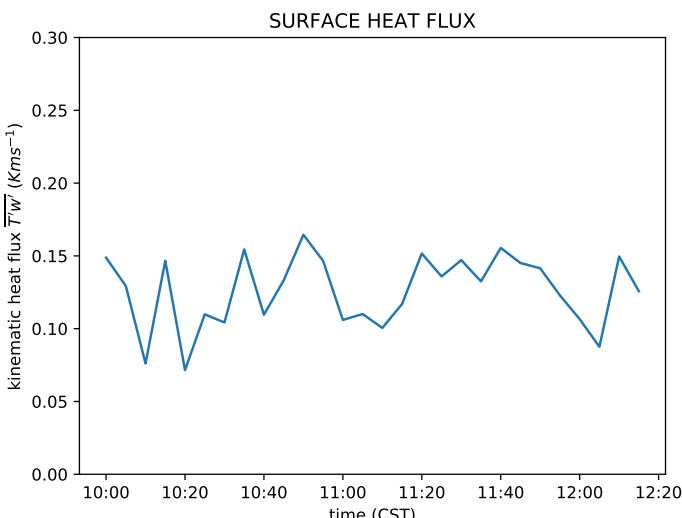

**Figure 2.** Five-minute averaged kinematic surface heat flux $\overline{T'w'}$ derived from 1 Hz wind and temperature sonic anemometer time series of the background ambient environment.

To help trigger convection in a horizontally uniform initial domain a small temperature perturbation "bubble" was added (see namelist.input_spinup in the Supplementary Materials). With periodic boundary conditions, near-stationary turbulence spectrum was achieved within ∼40 min of run start. The well-mixed modeled BL continued to turn over and warm for a total of 2 h 23 min (10:00:00 CST–12:23:00 CST). Restart file generated at 12:23:00 CST was used as initial conditions for the main burn simulation (12:23:00 CST–13:12:00 CST), ensuring the fire was ignited into a well developed BL. Other key configuration details can be found in Table 1, as well as in the complete namelist initialization files provided as Supplementary Materials.

**Table 1.** Key parameters of numerical domain setup.

| Simulation Parameter | Value/Description |
| --- | --- |
| Model version | 24 May 2019 |
| | (git https://github.com/openwfm/wrf-fire/tree/ced5955b23cfa9bc0f937783c1c63ff7aa1bc2fa) |
| Horizontal grid spacing | 40 m |
| Domain size | 260 grids (east-west) × 350 grids (north-south) |
| Time step | 0.1 s |
| Model top | 3000 m AGL |
| Spinup timing | 10:00:00–12:23:00 CST (CST = UTC − 6 h) |
| Fire (restart) simulation timing | 12:23:00–13:12:00 CST |
| Sub-grid scale closure | 1.5 TKE (TKE = Turbulence kinetic energy) |
| Lateral boundary conditions | periodic |
| Surface physics | Monin–Obukhov similarity (sf_sfclay_physics = 1) |
| Land surface model | thermal diffusion (sf_surface_physics = 1) |
| Surface heat flux | 160 W m$^{-2}$ (tke_heat_flux = 0.13) |

Following the LES spin up, the northwestern half of the simulated L2G lot was ignited with four roughly parallel fire lines mimicking strip head fire method used during the real-life burn (Figure 1). During the campaign, the prescribed burn was ignited with drip torches attached to moving all-terrain vehicles (ATVs). Using GPS data from these vehicles (available from [15]), we extracted the locations of start and end points of the four fire lines, as well as their individual start and end ignition times. While the real-life ignition process was not perfectly uniform in time, the modeled fire lines were approximated as being ignited at a constant speed, such that the time and location of the start and end points matched those of the real burn (see Animation S1 in the Supplementary Materials). Timing varied slightly for each of the four modeled fire lines (see namelist.input_main in the Supplementary Materials). We approximated the ignitions as straight lines between observed start and end points,

as the ATVs' deflections from a straight path during the real burn remained within a single atmospheric grid in our modeled domain.

Ignited cells in WRF-SFIRE proceeded to spread, while each fire line continued to advance until reaching the opposite end of the L2G lot. Subsequent upwind ignitions of the remaining lot area were excluded to reduce the computational load of the simulation. Taking into account the downwind location and timing of smoke plume observations, this simplification should have no effect on the proposed evaluation. The simulation was allowed to proceed for 49 min, until the emissions reached the downwind end of the domain.

Summary of fire and fuel parameters can be found in Table 2. Based on photographs and average measurements of fuel size, composition and type, we determined Anderson's fuel Category 1 (short grass) [19] to be the best fit for L2G ground cover. Actual burn perimeters were used to mask the remaining domain as containing no fuel to prevent spread of the simulated burn outside of the burn lot. We replaced the standard fuel loading and depth associated with Type 1 fuels with average measured values of 0.267 kg m$^{-2}$ and 0.18 m, respectively. Surface dead fuel moisture content was set to 8.46% based on observations. Heat of combustion of dry fuel was adjusted to $1.64 \times 10^7$ J kg$^{-1}$ as per estimates for grasslands provided by Overholt et al. [20].

**Table 2.** Details of fire and ignition parameters in LES setup.

| Simulation Parameter | Value |
| --- | --- |
| Fire mesh refinement | 10 |
| Ignition duration | 12:23–12:36 CST (varied for each fire line) |
| Rate of spread during ignition | 0.2 m s$^{-1}$ |
| Fuel category | 1 (short grass) |
| Surface dead fuel moisture | 8.46% |
| Heat of combustion of dry fuel | $1.64 \times 10^7$ J kg$^{-1}$ |

As the central goal of this work is to evaluate the model's ability to capture wildfire smoke plume dynamics, we did not incorporate chemistry coupling into the simulation. Modeled "smoke plume" was represented by two passive tracers released proportionally to the mass and type of fuel burned. The rate of release for each tracer representing CO and $CO_2$ was controlled by assigned emission factors, based on values for grasslands provided by [21] (see namelist.fire_emissions in the Supplementary Materials).

## 3. Results

The overall evolution of the simulated L2G burn and the associated smoke plume is best visualized with a 3D animation (see Animation S1 in the Supplementary Materials). The Supplementary Materials also includes an animated view of the cross-wind modeled $CO_2$ mixing ratio (Animation S2). The latter demonstrates the ability of the LES to capture common plume behavior. As seen in the animation, the initial rise of moist buoyant air results in a temporary overshoot of the equilibrium plume height, followed by the gradual settling of the plume to its final injection height near the top of the boundary layer for this case. While the ability of WRF-SFIRE to qualitatively capture typical plume dynamics is reassuring, the following sections take a more quantitative approach to model evaluation.

### 3.1. Fire Behavior

Prior to evaluating the ability of WRF-SFIRE to capture plume rise and dispersion, it is important to ensure that the model is able to reasonably simulate fire behavior. Initial surface and fuel conditions have the potential to strongly impact fire growth and intensity, and, hence, affect the location and buoyancy of the smoke plume. As noted in Section 1, our approach does not constitute a comprehensive fire behavior evaluation study, but rather aims to ensure that WRF-SFIRE captures the bulk properties of combustion and supplies a reasonable surface forcing to the simulated atmosphere.

Our evaluation is based on the analysis of fire energy transport of RxCADRE observational data for L2G burn carried out by Butler et al. [10]. The study provides measurement-based values as well as error margins for ROS, and peak and average heat fluxes of the fire, which we use to assess the performance of the semi-empirical fire algorithm driving our LES simulation. Figure 3a,b compares LES-derived average and peak total heat fluxes for HIP1 and entire burn area over the flaming period with observations. For HIP1 point-to-point comparison, we use output from the nearest modeled grid points. L2G average observed values include measurements from all three HIP lots. The corresponding simulated estimates are calculated using the entire burn area (roughly half of the L2G lot).

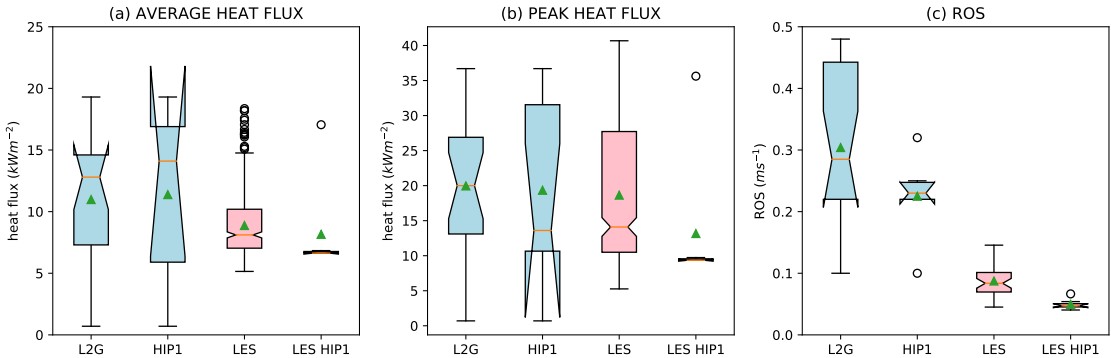

**Figure 3.** Comparison of observed (blue) and modeled (red) fire behavior. The box and whiskers span interquartile range (IQR) and $1.5 \times$ IQR, respectively, with the notch denoting the 95% confidence interval of the median (median $\pm 1.57 \times \text{IQR}/n^{1/2}$). Red line and green triangle correspond to median and mean, respectively. (**a**) Average heat flux during flaming period. (**b**) Peak fire heat flux during flaming period. (**c**) Rate of spread.

The start and end times of the flaming period are defined as simulation frames at which total heat flux at the location exceeded 5 kW m$^{-2}$ [10]. For both burn-wide and point comparisons, the flaming period is determined separately for each individual grid point. Only ignited grids are included in the analysis. This approach allows us to mimic the analysis performed by Butler et al. [10] in the absence of true combustion modeling in WRF-SFIRE.

For the entire burn area the observed mean and peak heat fluxes associated with the fire (not the background environment) are 11 kW m$^{-2}$ and 20 kW m$^{-2}$, compared to LES-derived values of 8.9 kW m$^{-2}$ and 19 kW m$^{-2}$, respectively. For HIP1 lot the corresponding values were 11.4 kW m$^{-2}$ and 19.4 kW m$^{-2}$ (observed) versus 8.2 kW m$^{-2}$ and 13 kW m$^{-2}$ (modeled). Note that, due to close proximity of the HIP1 sensors to each other, four out of seven of them fall into the same atmospheric grid within the modeled domain. Modeled HIP1 averages should therefore be treated with caution, as they consist of only four unique values. Moreover, the large spread of observed HIP1 heat fluxes renders the differences between model and measurements not statistically significant. Overall, the results shown in Figure 3 suggest that on average the surface thermal forcing to the modeled atmosphere due to the fire is reasonably captured by the model, subject to a slight negative bias (significant and non-significant for average and peak heat fluxes, respectively).

Observed rates of spread during the L2G burn were estimated using two methods in the study by Butler et al. [10]: flame arrival time from ignition and video images. The former approach takes into account the ignition time of the nearest fire line (perpendicular to fire advance vector) and the distance to the individual HIP1 sensors. The resultant values appear to have lower associated uncertainty than the latter image-derived method. To ensure consistency, we mimicked the above methodology in our simulated domain. Using the high-resolution fire domain, we calculated the upwind distance between each HIP1 point and the ignition line and the time it took the flame to reach each sensor location. To estimate ROS for the entire burn area, we created a mid-fire cross-section of 50 point-pairs between second and third ignition lines. Similar to the approach above, we derived the distance and flame

travel time for each pair to calculate ROS. As shown in Figure 3c, mean LES-based HIP1 and L2G ROS values of 0.049 m s$^{-1}$ and 0.087 m s$^{-1}$ are significantly lower then the corresponding observed rates of spread (0.23 m s$^{-1}$ and 0.30 m s$^{-1}$, respectively). Possible implications and sensitivity of our results to this deficiency are addressed in Section 4.

### 3.2. Plume Dynamics

Airborne emissions data collected during RxCADRE campaign is central to our evaluation of WRF-SFIRE's ability to capture plume rise and dispersion. The emissions dataset [17] contains smoke plume entry and exit points along the flight path, which were calculated using background CO baseline concentrations. The measurements were taken along horizontal transects passing through the plume at various vertical levels ("parking garage" profile), beginning close to the ground and moving towards the top of the plume, for a total of 9 crossings.

The identified in-plume segments were then compared with modeled CO mixing ratios along the same flight path extracted from the geo- and time-referenced LES domain. Figure 4 shows the time series of the flight path simulated emissions, overlaid with observations-derived plume segments. The results suggest good overall agreement in both location and timing between the modeled and observed emissions dispersion throughout majority of the BL depth. The coinciding model CO peaks and observed smoke segments indicate that the horizontal width of the smoke plume is well represented in the model. Potential shortcomings include excess smoke near the ground, as suggested by the early peaks (12:36 and 12:40 CST) not identified as a plume crossing, as well as a slight skew of the overall smoke distribution towards higher levels. A small phase shift appears in the modeled peaks toward the later parts of the simulation (12:50 CST and beyond).

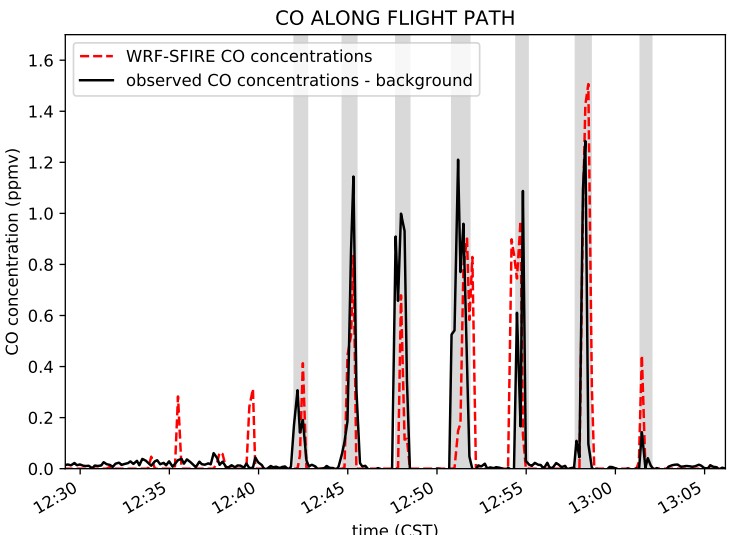

**Figure 4.** Simulated CO mixing ratio along RxCADRE flight path. Red dashed and solid black lines correspond to LES-derived and observed values, respectively. Gray shading indicates observed smoke time periods (not magnitudes) as identified from CO measurements along the flight path.

To evaluate the vertical distribution of WRF-SFIRE emissions, we compared the model-generated $CO_2$ concentrations with airborne measurements obtained during the "parking garage" and "corkscrew" (spiral ascent or descent) maneuvers. As shown in Figure 5a, there is a good overall agreement in injection heights for fire-generated emissions during the earlier "parking garage" profile. Plume top is accurately captured. Modeled concentrations tend to have a negative bias of ∼5 ppmv throughout the bulk of the plume thickness (500–1300 m), and be slightly over-predicted for the very top and bottom of the smoke column (at 400 m and 1500 m).

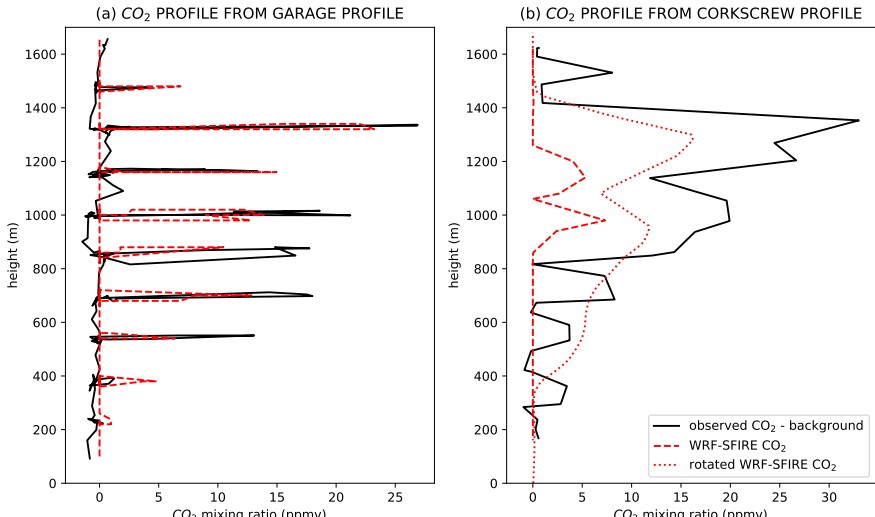

**Figure 5.** Observed (black) and modeled (red) vertical $CO_2$ emissions distribution during: (**a**) "parking garage" maneuver; and (**b**) corkscrew maneuver.

The "corkscrew" profile corresponds to a time near the very end of our simulation. As shown in Figure 5b, the band of modeled emissions appears to be very narrow and severely under-predicts the smoke concentrations. We discuss possible reasons for this behavior in Section 4.

## 4. Discussion

The aim of our WRF-SFIRE evaluation was to assess its ability to capture fire-generated emissions in the context of air quality. Hence, we examined the implications of the above results based on their potential applications for wildfire smoke plume rise and dispersion modeling. The following sections discuss model performance and accuracy from the perspective of atmospheric dynamics, as well as address potential implications of uncertainty in fire behavior and the associated input parameters.

### 4.1. Vertical Plume Rise in the Boundary Layer

As demonstrated in our results summary in Section 3.2, initially WRF-SFIRE produced a fairly accurate near-source emissions distribution and plume top with a slight under-prediction of concentrations (Figure 5a).

Over time model performance appears to deteriorate. Given that the fire thermal forcing compares relatively well with observations (Section 3.1), a more likely cause for the increasing difference between model and observations is background boundary layer dynamics. The atmosphere was initiated with 10:00 CST sounding, and continually forced with an observations-based constant surface heat flux. However, the cyclic lateral boundary conditions maintained the same vertical wind profile as initially supplied by the sounding at 10:00 CST, irrespective of potentially changing mesoscale conditions in the real atmosphere. Over the course of more than three hours between spin up start and the final minutes of the fire simulation, from which the corkscrew emissions distribution was obtained (Figure 5b), the real atmospheric wind profile likely evolved.

With time and further downwind the effects of any small changes in mesoscale conditions become more pronounced, which is why initially encouraging model performance deteriorated towards the end of the simulation. The markedly narrow band of emissions in Figure 5b suggests that the "corkscrew" location in the LES domain corresponded to the very edge of the plume rather than the center, indicating a shift in mesoscale wind conditions.

Indeed, analysis of observed background 30 m wind direction leading up to and during the burn shows a significant shift to the west, resulting in the LES "corkscrew" profile being extracted from the edge of the plume, rather then the intended center (Figure 6). Accounting for this observed wind

rotation, it is possible to extract a wind-corrected profile, such as shown with a red dotted line in Figure 5b. Assuming an average 20 degree rotation over the course of available wind observations (based on the slope of linear regression shown Figure 6a), the corrected location of the corkscrew maneuver indeed corresponds to the center of the plume (Figure 6b). The wind-corrected profile shown in Figure 5b is a notable improvement from the original non-rotated estimate. Note that this adjustment is extremely crude, as it is based on an estimated wind rotation at one point on a single vertical level and does not take into account potential changes in vertical wind shear.

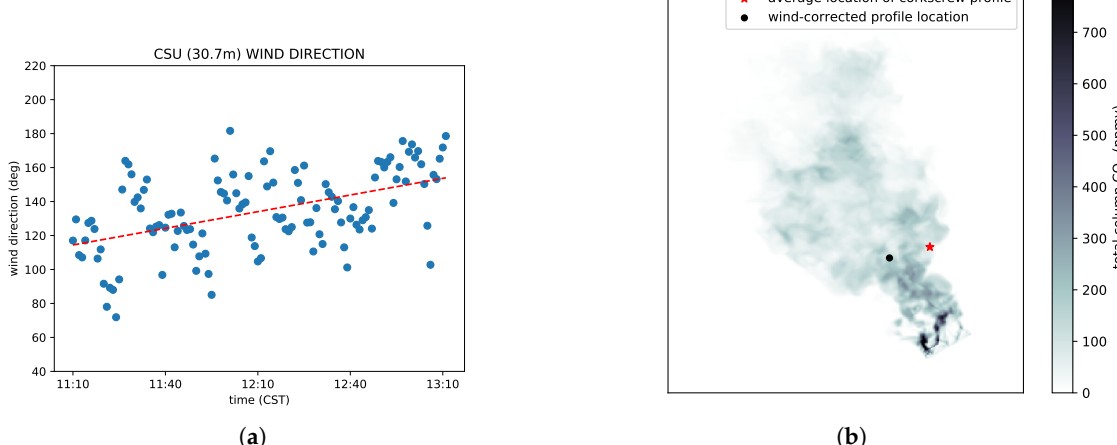

(**a**)  (**b**)

**Figure 6.** The effects of changing mesoscale wind conditions on plume observations (**a**) Observed change in 30 m wind direction prior to and during the burn. Significant linear trend is shown with a red dashed line. (**b**) Top view of modeled smoke plume during the "corkscrew" maneuver by the instrumented aircraft. Black dot and red star indicate the average location of the "corkscrew" profile from flight with and without wind-correction, respectively.

Unfortunately, unlike the Real-mode WRF simulations, there is no easy way to account for changing lateral boundary conditions in WRF-SFIRE large-eddy mode. Hence, we can expect the ability of the model to accurately capture dispersion to depend strongly on the variability of real background conditions as well as the simulation length and spatial extent of the modeled domain. Namely, an LES will provide better simulations for situations where that actual atmosphere is horizontally uniform and temporally steady. While this presents a limitation for smoke plume rise and dispersion modelers, it is important to consider it in the context of existing alternative sources of field data. Given a typical uncertainty of ∼500 m associated with the most accurate widely available plume height dataset from Multi-angle Image SpectroRadiometer (MISR) [22], WRF-SFIRE provides a valuable alternative source for generating comparatively accurate "synthetic plume height data".

Moreover, unlike instantaneous observational point measurements or overpass-limited derived satellite data, the LES allows us to examine the domain-wide temporal evolution of the plume and identify key features, which are likely to be of interest to dispersion modelers. As shown in Figure 7 and Animation S2, the vertical distribution of emissions in the domain changes throughout the simulation. Following an initial overshoot and a period of active smoke production near the ground, most of the emissions rise and end up near the top of the BL, accumulating just under the inversion level in a wide span of heights. While this vertical distribution may contain modeling and initial condition biases, it is likely to offer dispersion modelers an advantage over the common current approach of using a single empirically derived injection height.

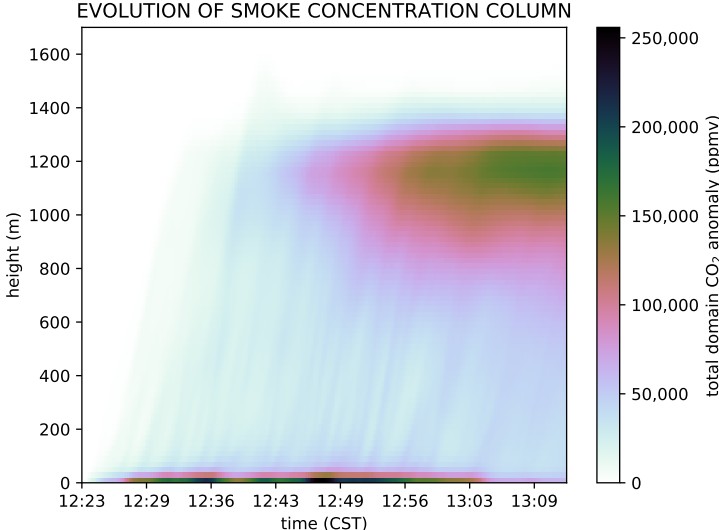

**Figure 7.** Evolution of total column $CO_2$ anomaly.

### 4.2. Importance of Fire Input Parameters

As noted in the Introduction, our evaluation work focused on assessing the relationship between coupled surface forcing and the atmosphere in WRF-SFIRE rather than on fire behavior. However, as we attempt to move forward from simple uncoupled burner-type experiments with prescribed constant surface heat flux to more realistic dynamic simulations, we must address the challenges in selecting proper fire input parameters.

Similar to Kochanski et al. [5], we found that the fire behavior model is particularly sensitive to the choice of fuel moisture. This parameter in WRF-SFIRE does not depend on the selected fuel category and was based entirely on measurements in our simulation. We also modified the standard fuel depth and loading parameters associated with Category 1 fuels to match observations, which resulted in very accurate surface heat flux forcing but substantially lower ROS values than observed or those obtained with standard settings.

Notably, similar thermal forcing to the atmosphere can be produced using a range of combinations of fuel categories and parameters in the model. We have not carried out a formal sensitivity analysis as it was beyond our scope and computational abilities, however, future modelers may find the following information helpful. As preliminary tests for our study, we have used Category 1 and Category 3 fuels (short and tall grass) with various combinations of both standard and measurement-based fuel depth and loading parameters to achieve similar surface forcing. The relationships between these parameters are highly non-linear, which makes determining the "correct" choice (in the absence of detailed observational data) difficult. What we found to be encouraging is that while the absolute value of modeled concentrations and ROS changes dramatically depending on the chosen fuel category for a given fire intensity, the relative distribution of emissions does not. The simulated atmosphere is forced solely by the parameterized heat and moisture fluxes, so WRF-SFIRE does not discriminate which combination of fuel characteristics produced a given heat flux that drives the buoyant plume rise.

Given any thermal forcing, the atmospheric response appears to be fairly robust, irrespective of the particular combination of fuel parameters or ROS with which it was achieved. While this study does not aim to establish whether the model sensitivity to fuel conditions is physical, it does suggest that the LES produces realistic plume rise for the given fire intensity.

### 4.3. ROS and Biases in Modeled Emissions

The model's poor performance for ROS in our case study likely resulted in reduced simulated emissions concentrations due to lower parameterized fuel consumption rate. This is consistent with the notable negative bias in our modeled $CO_2$ profiles.

As mentioned above, the low ROS values on our simulation are largely a result of our use of non-standard fuel depth and loading parameters. To eliminate alternative causes for slow fire line advance, we compared horizontal winds at the first and second model levels (at ∼8 m and ∼25 m AGL) with data obtained from 2D sonic anemometers mounted at multiple heights of the CSU-MAPS meteorological tower. As shown in Figure 8, the near-surface winds are generally accurately captured by the model. At the lowest vertical level, there tends to be a slight positive bias, which one would expect to contribute to higher rather than lower ROS values.

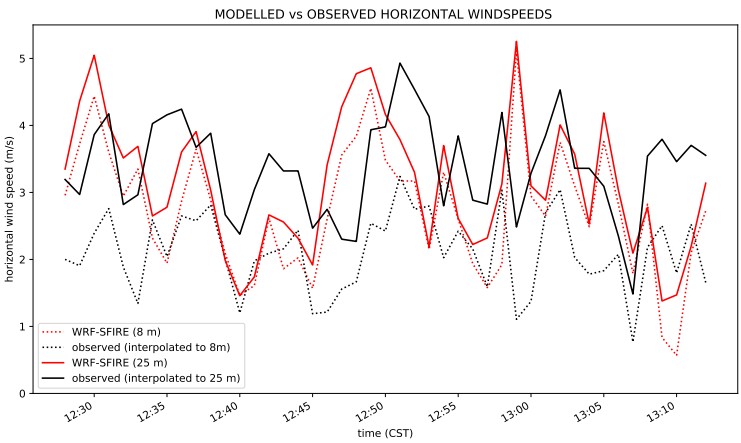

**Figure 8.** Modeled (red) and observed (black) near-surface horizontal wind.

Apart from their dependency on ROS and fuel consumption, the absolute values of WRF-SFIRE emissions are also controlled by user-prescribed emission factors. In our case study, these factors were not derived from measurements, but were rather based on standard values typical for the Grassland fuel category (see Section 2.2). Hence, the negative bias in our modeled smoke distribution could potentially be reduced, should observations-based emissions factors become available.

### 4.4. Experimental Design Considerations

One of the shortcomings of the RxCADRE dataset and this experiment is the substantial (nearly 2.5 h) difference in timing between the sounding balloon launch and the fire ignition. Availability of an additional vertical profile for model evaluation just prior to ignition would have been extremely helpful in mitigating some of the sources of error mentioned in the above sections. A similar recommendation was offered by Kochanski et al. [4], who suggested that an on-site sounding just prior to the burn rather than a few hours earlier would be most useful.

While we recognize the challenges of coordinating balloon launches in the presence of aircraft over the fire, a potential alternative would be to include on-board temperature and wind sensor data from flight with the smoke dispersion measurements.

### 4.5. Limitations

Recent studies suggest that the heat extinction depth parameter in WRF-SFIRE (or e-folding distance) has a strong influence on the modeled fire and near surface plume behavior [4,23]. Currently, there is no clear theory on how the vertical distribution of fire-released heat above the ground affects near-ground temperatures as well as ROS in the literature. As the relationship appears to be highly non-linear, we have not examined its implications in our simulations.

Overall, our findings suggest that the ability of WRF-SFIRE to capture plume dynamics of a specific real fire largely depends on the availability of timely atmospheric initial conditions and accurate simulation of fire intensity. Owing to the detail and comprehensive nature of the data provided by the RxCADRE experiment, these critical inputs could generally be derived from measurements for the current case study. This sensitivity, however, could present a challenge for future real-time fire simulations, where few or no such measurements would be available.

## 5. Conclusions

This work aimed to assess the ability of a coupled fire–atmosphere WRF-SFIRE LES model to simulate a case study of fire smoke plume growth and dispersion. We examined the L2G burn from the RxCADRE 2012 campaign—a comprehensive experiment combining simultaneous monitoring of fuel, fire behavior, meteorology and emissions.

Our model evaluation demonstrates good overall agreement between the LES and the observations, subject to accuracy and timeliness of model initialization data. Using the emissions and dispersion data collected from an airborne platform during the RxCADRE experiment, we show that LES reasonably captures the timing, rise and dispersion of the fire plume. We examined the possible relationships among model biases, fire behavior and changes in ambient atmospheric conditions.

The work demonstrates the utility of WRF-SFIRE LES in studying some aspects of fire plume dynamics. The scarcity of detailed plume observations presents one of the central challenges for smoke-model development. WRF-SFIRE's ability to capture the rise and spread of fire emissions for cases such as studied here has the potential to address this critical research need and provide alternative "synthetic" data for future development of parameterizations for wildfire smoke plume rise.

**Supplementary Materials:** The following are available online at http://www.mdpi.com/2073-4433/10/10/5 79/s1, Animation S1: WRF-SFIRE simulated fire and smoke over real terrain. Visualization produced using VAPOR software. Animation S2: Cross section of WRF-SFIRE simulated emissions along mean wind direction. WRF-SFIRE_init_files.zip: All files required to initialize and run the model simulation.

**Author Contributions:** Conceptualization, N.M. and R.S.; Formal analysis, N.M.; Funding acquisition, R.S.; Methodology, N.M.; Resources, R.S.; Supervision, R.S.; Visualization, N.M.; Writing—original draft, N.M.; and Writing—review and editing, R.S.

**Funding:** This work was funded by grants from Natural Sciences and Engineering Research Council of Canada (NSERC) and BC Clean Air Research Fund (CLEAR).

**Acknowledgments:** The authors would like to acknowledge Brian Potter, Ronan Paugam, Ruddy Mell, Derek McNaran and Adam Kochanski for their input and collaboration. Thanks are also given to Daisuke Seto and Craig Clements for their help with obtaining RxCADRE sounding data and the UBC Weather Research and Forecasting Team for their ongoing support.

**Conflicts of Interest:** The authors declare no conflict of interest.

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
