# Peer review of "Capturing Plume Rise and Dispersion with a Coupled Large-Eddy Simulation: Case Study of a Prescribed Burn"

_atmosphere, doi:10.3390/atmos10100579_

Round 1

Reviewer 1 Report

The authors described the use of tracers and the ignition process, as requested in the previous review.

They still need to identify the versions of the software and the settings used for reproducibility of the results.

Author Response

Please use Adobe Acrobat Reader to ensure attachments embedded within the PDF file are visible. 

Reviewer 2 Report

General comments:
The authors did a great job in improving the manuscript. I like the new version of the manuscript and I appreciate authors effort put into addressing reviewers' comments. Unfortunately, there is still one very significant issue that has to be addressed before the paper can be accepted for publication.

Trying to address the problem with the missing turbulence spin-up the authors took advantage of the WRF functionality that implements a so-called warm/cold bubble. What this functionality does, is that it modifies the air temperature within a specific volume of suspended in the air in order to initialize localized strong convection or to generate a downburst or a density current. This functionality is not intended to spin-up turbulent eddies, but to induce a very local positive/negative buoyancy. In this particular study, the introduction of this perturbation (of a 50m radius) can significantly alter the simulated vertical velocities and the vertical smoke distribution by introducing an artificial (+10K) buoyant forcing that generates an additional superficial updraft not related to the fire itself. I know the authors would like to have this work published as soon as possible and I would like too, but I have to request rerunning the simulations with the warm bubble deactivated as the current configuration, can't be treated as valid modeling setup for the assessment of the simulated plume dynamics.

What should be done instead, is introducing a small (in the order of +/- 1.5K) randomized perturbation in the surface temperature that won't change the domain averaged surface temperature but will introduce a thermal inhomogeneity that will promote the turbulence spin-up.

Also, in order to carry on the fully evolved turbulent simulation, the 'main' simulation should be run from a restart file generated at the of the spin-up simulation, not the output. A run stated from the output as suggested in Line 107, would be still a cold-start simulation which is not what the authors should do trying to preserve the turbulent characteristics. I hope that is just a typo in the text, as the namelist files suggest that the main run was initialized properly from the restart file.

Detailed comments:
L274, Figure 7 according to its caption still implies that that the water vapor is shown, while the color bar legend suggests that it is CO2. The authors used two traces (one for Co, and one for C02), so this plot should present the actual simulated CO2 concentrations, not the water vapor. I hope it is a captioning problem.

Author Response

Please use Adobe Acrobat Reader to ensure attachments embedded within the PDF file are visible. 

This manuscript is a resubmission of an earlier submission. The following is a list of the peer review reports and author responses from that submission.

Round 1

Reviewer 1 Report

The paper is well written and quite clear. However, much crucial information is missing and significant improvements are in order.

The files essential for reproducing the simulation must be attached; in particular, at least namelist.input and namelist.fire, and also any files read  by ideal.exe need to be submitted with supplementary materials

Include information what version (git comit hash and date) of the software the authors used. A description of changes made to the code and any modified source files should be included in supplementary materials.  

Water vapor transfer is of interest and can be kept but it is really not a good proxy for smoke. There are smoke tracers in the software for that purpose and the authors should use them.

The ignition process used in the software is not described adequately. Implementation of the ignition  must be described verbally in the paper as well as be reproducible from the supplementary materials requested above.

Reviewer 2 Report

The presented work is an attempt to asses the capability of a coupled fire-atmosphere model in terms of rendering plume rise and dispersion, based on the RxCADRE experimental data. The paper is clearly organized and well written. Its subject is definitely significant from the standpoint of advancing current capabilities in terms of modeling plume rise and dispersion.  Unfortunately, serious flaws in the experimental design and description of the method preclude it from publication in its current form. The results discussion is currently rather week and not well supported due to the shortcomings of the experimental setup and insufficient validation of the atmospheric component of the system.  

Major comments:

1.  Using the water vapor as a tracer, and pretending that the atmosphere is dry is not acceptable. The model supports a passive tracer which should be used for the purpose of the smoke characterization. By assigning the tracer emission factors to the observed CO emission factors, the CO concentrations could be computed directly in the model enabling much more informative, direct comparison between the model and aircraft data.  The good thing that the authors have all the tools to do that in hand. No chemistry is needed for that, just tracer_opt=2 and custom entry in namelist .fire_emissions.

2. The general problem of the simulation used in this study is the fact that it doesn't seem to be initialized properly. As the model is run in the LES mode, proper evolution of eddies is crucial for the development of the realistic convective boundary layer.  Authors commented on the probable with too shallow boundary layer but didn't connect it to the flaws in the modeling setup.

The tke_heat_flux used in this study is not the way to go as it is uniform for the whole domain, and in conjunction with the typical uniform frictionless surface used in idealized cases, there is no mechanism to spin up the eddies and promote realistic PBL growth. That in turns translates into problems with the vertical plume extent. Again, the right tools for that are available. The surface_init option for idealized cases enables proper surface initialization in terms of the surface type, soil temperature, and moisture as well as near-surface temperature. A random perturbation of the initial surface temperature routinely used in LES simulations would enable eddy formation and would result in improved representation of the PBL growth. 

3. It is hard to diagnose the rate of spread not knowing how well the model simulated local winds. The RxCADRE L2G burn was equipped with multiple anemometers, a main meteorological tower and a mobile tower measuring winds at multiple levels. The authors should take advantage of these measurements in the very first step of their validation. Also, the sonic anemometers within the plot could provide information about the in-plume vertical velocities that should be carefully examined to evaluate the simulated plume rise. In the presence of an inversion layer, the vertical plume extent is linked with the model ability to resolve the inversion rather than the actual plume dynamics.

4. In the current version of the paper (mostly due to the lack of the namelists that supposed to a part of the appendix, but are missing), the description of the modeling setup is insufficient. For example, what type of boundary conditions, the spacing between the vertical levels, vertical wind, temperature and moisture profiles used for the initialization, the ignition procedure, type and properties of the ground layer (land use type, temperature, etc.)  are described with satisfactory details. Also, it would be great to know what version fo the code authors were using and if any code modifications were made for the purpose of the presented simulation.

Detailed comments:

L 33. Large-scale plume tops simulated with WRF-SFIRE (RANS) were evaluated based on MISR data in Kochanski, et, al. 2015 (http://dx.doi.org/10.1071/WF14074) it would be useful to add this reference and point out the uniqueness of this study which focussed on small-scale LES simulations and utilizes aircraft data.

L 94. Please clarify, the latent heat flux in WRF-SFIRE is computed based on the fuel consumption, and stoichiometric combustion of cellulose. So the water vapor flux from the fuel thermal decomposition should be accounted for.

L104-108 please see comment 2, and 4. More details are needed here.

L116-119, the ignition procedure needs more description. From the animation is look slike all the ignition lines reached the edge of the experimental plot perfectly at the same time, while during the actual burn it wasn't the case. Please explain how the ignition lines were implemented. Based on the actual GPS and time data, or just approximated by straight lines.

L154. Please increase the size of Figure 3. In the current form, it is hard to read.

L177-178 How well the simulated winds compared to the observations?

L 226-227. That is not entirely true because instead of passive CO, the authors used water vapor that is not passive in the atmosphere.  See comment 1.

L241-246. How the turbulence was initialized in the model? With the enforced surface heat flux insufficient evolution of the eddies could be potentially to blame. See comment 2.
